# What Exactly Is Inflammation (and What Is It Not?) [note 1]

**DOI:** 10.3390/ijms232314905

**Published:** 2022-11-28

**Authors:** Bryan Oronsky, Scott Caroen, Tony Reid

**Affiliations:** EpicentRx Torrey Pines California, La Jolla, CA 92037, USA

**Keywords:** inflammation, PAMPs, DAMPs, LAMPs, SAMPs, immune interventions, innate immune cells

## Abstract

In medicine, inflammation is a fuzzy, overused word first coined by the Romans, the intended meaning and precise definition of which varies according to the person and the clinical context. It tends to carry a negative connotation as a response gone awry, like a raging, out-of-control wildfire that requires immediate control and containment lest it destroy all in its path; however, frequently overlooked or lost in the shuffle is the primordial importance of inflammation to health and survival. The precise definition of inflammation matters for several reasons, not least because of the over-liberal use of anti-inflammatory drugs to inhibit inflammation, which may, contrary to prevailing dogma that all inflammation is harmful, act counterproductively to prevent restitutio ad integrum. Using fire as a central analogy, this overview attempts to define inflammation, the better to determine how to manage it, i.e., whether to fan its flames, let it burn out, or suppress it entirely.

## 1. Introduction—An Analogy with Fire

Inflammation is an age-old, ancestral word, which comes from the Latin inflammare, meaning to ignite or burn. An analogy with fire is instructive because teleologically the purpose of inflammation is to pre-emptively ‘ignite’ in defense against an array of potential threats and then to spontaneously extinguish after threat neutralization [1]. The problem with inflammation is three-fold: (1) not all threats necessarily warrant an inflammatory response, including blunt trauma, ischemia-reperfusion injury, exposure to toxins or crystal particulates and auto-inflammatory diseases; (2) inflammation is an “equal opportunity offender” that ‘singes’ both diseased and healthy tissues; and (3) as with any fire, the potential for smoldering persistence or uncontained inflammatory spread is ever-present. The authors wish to emphasize that this is a perspective, or a point of view on inflammation, not an in-depth review, which can be found elsewhere [2], and, as such, it only touches the surface of what is known, understood, or theorized about inflammation. Its intent is to broadly define and contextualize what inflammation is (and, also, what inflammation is not), with the aim being to better manage it.

## 2. The First Clinical Definition of Inflammation—A Fire Alarm

Aulus Cornelius Celsus (25 BC-50 AD), a Roman encyclopedist, first itemized the four cardinal signs of inflammation: rubor et tumor cum calore et dolore, that is, redness and swelling with heat and pain, which metaphorically overlaps with the red-hot, burning pain from fire [3]. In the 19th century, Virchow promulgated a fifth sign, that of the non-assonant functio laesa or loss of function [4].

Despite having stood the test of time and albeit still in wide medical use today, this clinical definition is overly simplistic and potentially misleading, as (1) it does not address the pathophysiology of inflammation; (2) implies harm from a process (because fire is associated with danger) which is at its core protective and whose benefits often far outweigh the risks; and (3) the five signs of rubor (from vasodilation), tumor (from exudation), calor (from hyperemia), dolor (from local tension and inflammatory mediators), and functio laesa (from loss of mobility or fibroplasia) are rarely present in all cases, especially when inflammation is systemic rather than localized to the tissues in one area.

## 3. Modern Definition of Inflammation—Good Fire vs. Bad Fire

With the benefit of insights from molecular biology that were unavailable before the 1960s, it is possible to define inflammation more broadly as a protective response, involving the activation of immune and non-immune cells, in response to an insult such as infection, toxic compounds, damaged cells, or irradiation, with the aim to restore tissue homeostasis. This raises the question as to whether inflammation is always protective [5]. The answer is a less-than-definitive, “it depends”.

Like a controlled burn or “good fire”, which is confined to a prescribed area, acute inflammation exists, helpfully, to clear away unwanted debris, i.e., pathogens, or damaged tissue in service of repair, regrowth, and regeneration. Ideally, the resolution of inflammation and a return to baseline status in days to weeks follows the eradication of the inflammatory stimulus. Alternatively, inflammation is harmful when it fails to resolve because of the persistence or propagation of the noxious inflammatory stimuli. In this case, the controlled burn of inflammation may either (1) smolder, i.e., turn chronic, lasting for days or years, leading to fibrosis and/or dysfunction (Figure 1); or (2) accelerate and spread like wildfire to the extreme detriment of other tissues and organs.

Persistence or chronicity occurs when the inflammatory response is unequal to the task of eliminating antigenic stimuli, such as foreign bodies or microorganisms, or other inflammatory triggers including urate crystals, cholesterol crystals, silica, and excess reactive oxygen species (ROS) production [6]. The chronicity of inflammation is associated with a host of conditions, such as inflammatory bowel disease, cancer, type 2 diabetes, heart disease, and autoimmune disorders such as rheumatoid arthritis, multiple sclerosis (MS), and systemic lupus erythematosus (SLE). At the other end of the spectrum is hyperinflammation, which occurs with the exaggerated expression of pro-inflammatory mediators as in sepsis, septic shock, and severe COVID-19 [7].

## 4. The Arsonists and the Firefighters

### 4.1. The Arsonists—Neutrophils, Macrophages, T Cells, and Cytokines

In response to tissue damage from trauma, microbial invasion, or noxious chemicals, pre-stationed sentinel cells, such as macrophages and endothelial cells, and a complex of activated proteins called the inflammasome ‘call up’ the arsonists (i.e., neutrophils and monocytes, from which macrophages and dendritic cells are derived) to the inflammatory front lines. Neutrophils, the cell type that first ‘swarms’ to the site, phagocytose cellular debris and release lysozyme-, matrix metalloproteinase-, and myeloperoxidase-containing granules as well as reactive oxygen species (ROS) and cytokines, such as IL-1, IL-6, and TNF α, to kill microorganisms and degrade apoptotic bodies. Approximately 48 h later, a second cellular wave of infiltration of monocytes that differentiate into pro-inflammatory M1 macrophages arrive; their functions include the removal of the apoptotic bodies of dead neutrophils, a process termed efferocytosis, production of pro-inflammatory cytokines and lipids, such as eicosanoids, and antigen presentation to T lymphocytes, which lead to the production of antibodies, cytokines, and memory cells. In particular, activated T lymphocytes directly contract monocytes/macrophages, which triggers pro-inflammatory cytokine production [8].

### 4.2. The Firefighters—M2 Macrophages, Tregs, and Pro-Resolving Mediators

The process of resolution or catabasis and the return to homeostasis begins with the “clean up” and elimination of apoptotic neutrophils, foreign agents such as bacteria, and necrotic debris from the site by macrophages. This triggers the switch of pro-inflammatory (M1) macrophages to an anti-inflammatory and pro-resolving M2 phenotype. In addition to M2 macrophages, regulatory T (Treg) cells, which produce the anti-inflammatory cytokines IL-10 and TGF-β and scavenge IL-2 and myeloid-derived suppressor cells (MDSCs) that inhibit T cell function, contribute to the resolution process. A series of pro-resolving mediators that includes lipoxins, resolvins, protectins, maresins, proteins and peptides such as annexin A1 (AnxA1), galectins, adrenocorticotropic hormone (ACTH), gaseous mediators such as hydrogen sulfide (H_2_S) and carbon monoxide (CO), nucleotides such as adenosine, and neuromodulators, such as acetylcholine, are also involved.

An inadequate or insufficient resolution leads to prolonged and chronic inflammation, illustrated graphically in Figure 1, that is responsible for excessive tissue damage, fibrosis, and multiple disease states including autoimmunity [9].

## 5. Fire Accelerants—DAMPs, PAMPs and LAMPs

Fire accelerants are damage-associated molecular patterns (DAMPs), released from stressed or dying cells; pathogen-associated molecular patterns (PAMPs), which are common molecular structures found on pathogens or the cells they infect, such as lipopolysaccharide (LPS), peptidoglycan, and betaglucan; and the recently classified lifestyle-associated molecular patterns (LAMPs), which include cholesterol, monosodium urate, oxidized LDL, and other lifestyle-related molecules that induce a sterile, chronic inflammation. This is shown in Figure 2. DAMPs, PAMPs, and LAMPs attach to pattern recognition receptors (PRRs) expressed on/in innate immune system cells and the inflammasome complex of proteins, which recognizes them, to mediate an inflammatory response [10]. Eventually, in the presence of microbial, altered-self antigens or tumor antigens, an adaptive immune response develops [11].

## 6. Fire Retardants—SAMPs

A functional counterpoint to pathogen-associated molecular patterns (PAMPs) and damage-associated molecular patterns (DAMPs), conserved molecules that are indicators of pathogen invasion and host cellular damage, is the self-associated molecular patterns (SAMPs), such as prostaglandin E2 (PGE2), annexin A1 (AnxA1), and specialized pro-resolving mediators (SPMs), mentioned earlier, which downregulate immune responsiveness and decrease oxidative stress [12]. These SAMPs act in the capacity of suppressing/inhibiting DAMPs, which decreases the reactivity of innate immune cells following an immune response. In this regard, hyper-inflammation or chronic inflammation may result not only from excessive or prolonged production of DAMPs, but also impaired generation of SAMPs.

## 7. Management of Inflammation: Fan the Flames, Let It Take Its Course, Let It Burn, or Prevent It

### 7.1. Fan the Flames

Two cases in point illustrate where the resolution phase of inflammation is not always beneficial. These are advanced cancer and systemic inflammatory response syndrome (SIRS)/sepsis, both of which are initially associated with pro-inflammatory activation, but which subsequently give way to a reactive suppressing anti-inflammatory response. Indeed, for several tumor types, including melanoma and non-small cell lung cancer (NSCLC), one of the most effective treatment options are checkpoint inhibitors that antagonize this counter-reactive suppressive, anti-inflammatory response, which cancers use to evade elimination [13,14]. Checkpoint inhibitors are also under intensive preclinical and clinical investigation in sepsis to block the complex of hypo-inflammatory, counter-regulatory mechanisms, known collectively as the compensatory anti-inflammatory response syndrome, or CARS, that develops simultaneously. Furthermore, the use of checkpoint inhibitors may correct an inadequate adaptive immune response that in the first place is responsible for a persistent inflammatory state, and which, in turn, can lead to a cytokine storm, acute respiratory distress syndrome (ARDS), and organ failure. CARS leads to reduced antigen presentation, increased lymphocytic apoptosis, and decreased leukocyte recruitment of leukocytes, all of which increase the susceptibility to adverse outcomes from secondary infections [15].

### 7.2. Let It Flame Out, i.e., Let It Take Its Course or If in Doubt, Do Not Put It Out

Like fever, with which the inflammatory response is closely linked, it is not always helpful to suppress inflammation [16]. This makes sense from an evolutionary perspective, since the inflammatory response appeared two billion years ago, when the innate immune system emerged, well before the arrival of humans [17]; hence, the treatment of inflammation, particularly when wound healing and infection control are involved, may impair immune competence, and increase susceptibility to infection. The phrase “if in doubt, don’t put it out” refers to a “watchful waiting” or active surveillance period, deferring immediate treatment in order for a natural resolution to spontaneously occur prior to the start, at some time in the future, of non-antibiotic and antibiotic anti-inflammatory interventions, the overprescription and overuse of which is a major concern [18,19]. The watchful waiting period will vary according to the clinical context and patient perception. These treatment options may include non-steroidal anti-inflammatory drugs (NSAIDs) that inhibit cyclooxygenase (COX) enzymes, which convert arachidonic acid into the inflammatory metabolites, prostacyclin, prostaglandins, and thromboxane to treat, e.g., soft tissue injuries; corticosteroids, to treat, e.g., COPD; macrolides, which are bacteriostatic antibiotics such as azithromycin and clarithromycin, to treat, e.g., acute rhinosinusitis; and hydroxychloroquine, a the disease-modifying anti-rheumatic, which is a hydroxylated derivative of chloroquine, to treat, e.g., COVID-19. The reason to perhaps withhold or defer administration of these anti-inflammatory therapies, several of which, like corticosteroids repress the mRNA transcription of pro-inflammatory genes in macrophages and neutrophils, is their potential to interfere with the removal of the inflammatory stimulus. If this stimulus is not removed, persistent inflammation may result, which is responsible in the long run for excessive tissue damage and pathology.

### 7.3. Put It Out or Suppress It

Four major classes of immune-directed therapies are available to suppress inflammation, especially in allergic, autoimmune, and fibrotic diseases (See Table 1).

The first is antigen-directed immunotherapy, the classic example of which is allergen hypo-sensitization. In allergic diseases, such as asthma, allergic rhinitis, and atopic dermatitis, where well-defined antigens, such as dust mites, grass and weed pollen, pet dander, and mildew, are identified as causative, multiple courses of allergen delivery via subcutaneous or sublingual routes induce a switch from a “Th2-type” response, with dominant production of IL-4 and IL-5, in favor of a “Th1-type” response, with the production of interferon gamma and IL-2 [20]. An adequate treatment duration, often lasting several years, may render the desensitization effect permanent.

The second class of therapy is immunomodulation, with therapies including hydroxychloroquine (HCQ), or Plaquenil, an anti-malarial used in several autoimmune disorders, such as systemic lupus erythematosus (SLE), primary Sjögren’s syndrome, and rheumatoid arthritis (RA), which inhibits antigen presentation. In terms of its effect on the immune response, HCQ is thought to reduce the efficacy of antigen presentation [21]. Other antimicrobial immune modulators include sulfasalazine, which is primarily used for rheumatoid arthritis and ulcerative colitis (UC); dapsone, which is used for cutaneous lupus [22]; and macrolides, which are used for chronic obstructive pulmonary disease (COPD).

The third class of therapy is anti-fibrotic agents, of which only two are approved (nintedanib and pirfenidone) for the treatment of idiopathic pulmonary fibrosis (IPF). These therapies have been used as treatment options for post-COVID-19 and post-ARDS pulmonary fibrosis [23].

The fourth and “bluntest” class of therapy is the immunosuppressive agents. These include corticosteroids, such as prednisolone, hydrocortisone, and methylprednisolone, whose risks tend to outweigh the benefits if used over the long term [24], steroid sparing disease modifying anti-rheumatic drugs (DMARDs) and anti-cytokine therapies, such as methotrexate, adalimumab, abatacept, etanercept, rituximab, tocilizumab, anakinra, ustekinumab, secukinumab, tofacitinib, and baricitinib, which are useful in a range of systemic autoimmune conditions [25].

## 8. Prevent It

Given the potential for inflammation to spike uncontrollably or to turn chronic, prevention is a priority. The pillars of prevention to reduce pro-inflammatory activity are a healthy diet low in saturated fats and refined sugars and high in complex carbohydrates [26], fiber, protein, and polyunsaturated fatty acids (PUFAs); regular physical activity; low or no exposure to air pollution, environmental and industrial toxicants, and tobacco smoking; good sleep quality; social well-being; and minimal psychological stress [27,28]. These beneficial behaviors may lead to reduced visceral and ectopic fat accumulation, which is associated with systemic inflammation.

## 9. Discussion and Conclusions

This review has attempted to define inflammation in order to better manage it, using an extended comparison with fire throughout as an organizing and simplifying principle for a complex topic. Lest this comparison be criticized as heavy-handed and excessive, the authors are at pains to reiterate that the Ancient Romans, in their wisdom, handed down the word inflammare, meaning to set alight or to burn, over 2000 years ago, thus inextricably linking the two words, fire and inflammation, which are often juxtaposed in the scientific literature [1,29,30].

Inflammation is a commonly used word in medicine, whose meaning is variable, fuzzy, and inconsistent because no consensus definition has ever been arrived at, it is a catch-all term used to describe many situations, and even among physicians, inflammation is widely perceived as a pathological condition that requires suppression with anti-inflammatory therapies. This despite the fact that inflammation is actually a protective response on which survival critically depends, especially during injury and infection, and despite the association of anti-inflammatory therapy administration with worsened clinical outcomes in sepsis. Moreover, inflammation is not a static process, but a dynamic one that changes and evolves over time, usually to resolution and a return to tissue homeostasis, if left to run its course [31].

The fact that inflammation damages healthy and diseased tissues alike is a feature, not a bug: this is the price to pay for the wholesale elimination of microbial pathogens, which pose a dire, immediate threat to survival. On par with the aggressively antagonistic military motto, “Kill ‘Em All, Let G** Sort ‘Em Out”, the priority and emphasis of the inflammatory response is threat removal, not specificity or selectivity. Collateral damage is the inevitable byproduct of massive ROS and proteolytic enzyme generation from hyper-activated leukocytes [32].

The calculus is less clear-cut for sterile inflammation, the etiologic factors of which are irritant particles such as asbestos, cholesterol, monosodium urate, and amyloid-β; these particles often do not, in and of themselves, pose a threat, but their incomplete or “frustrated” phagocytosis by macrophages leads to chronic inflammatory injury [33].

Whatever the trigger for it, whether infectious or sterile, the acute inflammatory response involves an influx of cytolytically activated cells of the innate immune system and the release of inflammatory mediators that increase vascular relaxation, vascular permeability, and the extravasation of large molecules from the circulation to the site of tissue damage; this leads to the Celsian tetrad of dolor, calor, rubor, and tumor, as well as a loss of tissue and organ function in severe cases.

Where inflammation ‘goes off the rails’ is when it fails to eliminate the causative factor(s) and either becomes chronic, which contributes to a wide variety of disorders, such as cancer, type 2 diabetes, heart disease, and autoimmune diseases, or it ramps up uncontrollably, as in severe sepsis. According to Newton’s Third Law [34], every action is followed by an opposite reaction; in inflammation, this leads to counter-regulatory anti-inflammation and immunosuppression, the unintended consequences of which in conditions such as sepsis, cancer, trauma, ischemia-reperfusion injury, and burns are increased susceptibility to adverse outcomes. Paradoxically, this has led to treatment with immunostimulatory agents.

This leads back to the titular questions: what exactly is inflammation, and, by extension, what is it not? Inflammation is a physiological, tightly regulated, protective response to an underlying infectious or non-infectious process or condition that involves cells of the innate immune system, adaptive immune system, and inflammatory mediators; it is not a pathological response, despite the over 3000 disease conditions that end in “-itis”, a suffix which denotes or connotes inflammation [35]. What is often lost—or forgotten—is that inflammatory responses are beneficial, and biologically appropriate and necessary, if, as is usually the case, they self-terminate with the destruction of the injurious agents that initiated them and with the restoration of tissues to their pre-inflamed state. On the other hand, inflammatory responses are maladaptive and, therefore, harmful if they are self-directed, too extreme, or fail to resolve in a timely manner so that the pathology from persistent inflammation ensues.

Of the several options that exist to manage inflammation, prevention is better than treatment, to paraphrase Erasmus and Benjamin Franklin [36]. However, if prevention is not an option, then three potential targets for treatment are (1) the prime etiologic factor, (2) mediators of tissue injury, and (3) mediators of tissue resolution. In the case of option (1), this would involve the use of specific agents to alter or eliminate the causative factor, such as antibiotics, antifungals, and antivirals, for an infection caused by bacteria, fungi, or viruses, respectively, statins for hypercholesterolemia, allopurinol for hyperuricemia, surgery for appendicitis or cholecystitis, weight loss for obesity, proton pump inhibitors for gastroesophageal reflux (GERD), etc.

In the case of options (2) and (3), a dynamic analysis of inflammatory biomarkers may help to inform the clinical decision whether and how to treat, as shown below in Figure 3, since the cytokine profile may define and describe an ongoing inflammatory process in terms of the nature of the insult, infection, or injury. This analysis, which is routinely performed with the enzyme-linked immunosorbent assay (ELISA), involves the detection and quantification of one or more cytokines from a multitude of biomolecules present in any given sample. When sustained pro-inflammatory hypercytokinemia is present with, for example, IL-6, IL-1, and TNF-α, as in septic shock, cytokine storm, acute respiratory distress syndrome (ARDS), and cancer-related cachexia, the potential treatment options, which are controversial, include anti-cytokine antibodies, such as the IL-6 inhibitor, tocilizumab, the IL-1 receptor antagonist, anakinra, and the TNF-alpha inhibitor, infliximab, and extracorporeal blood purification to reduce excessive levels of PAMPs, DAMPs, cytokines, and other pro-inflammatory mediators [37].

When chronic inflammation is present, treatment with antigen-directed immunotherapy, immunomodulatory, anti-fibrotic and immunosuppressive agents is potentially warranted, depending on the clinical context. In combination with other pro-inflammatory, pro-fibrotic biomarkers, such as C-reactive protein, ferritin, serum amyloid A (SAA), pro-calcitonin and transforming growth factor-β (TGF-β, a key driver of fibrosis), cytokine profiling, since cytokines mediate or regulate the inflammatory process, may hold promise to determine (or to confirm) not only whether inflammation is present and to what degree, but also to decide whether treatment is warranted and, if so, to select the most appropriate treatment and to predict the response to it [38].

In summary, then, inflammation is an “old flame”, whose mostly beneficial, life-saving heat has been felt since the dawn of mankind and which requires careful tending, lest it burn too cold or too hot.

## Figures and Tables

**Figure 1 ijms-23-14905-f001:**
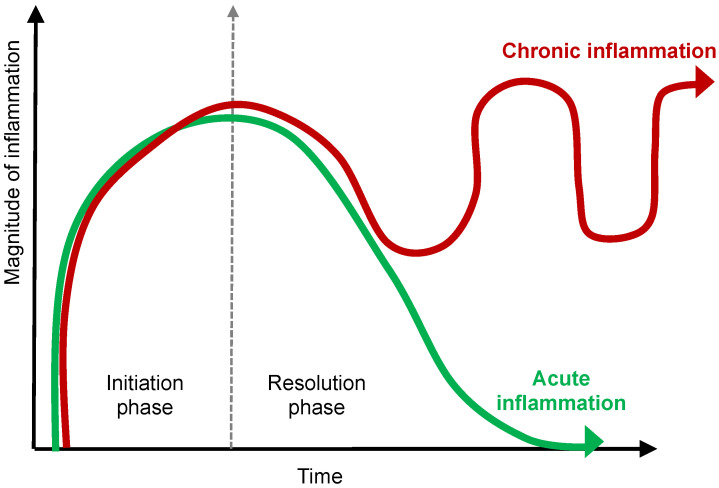
The Dynamics of Chronic Inflammation. The acute inflammatory response is transient and programmed to resolve, which is what occurs in a state of health. Failure of resolution results in prolonged or chronic inflammation with tissue remodeling, fibrosis, and disease.

**Figure 2 ijms-23-14905-f002:**
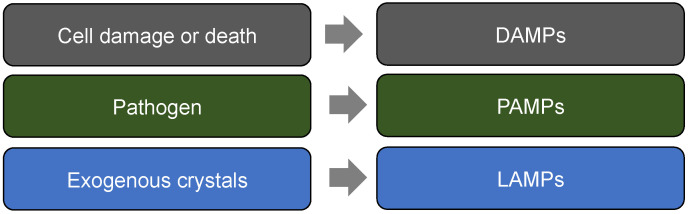
DAMPs vs. PAMPs vs. LAMPs.

**Figure 3 ijms-23-14905-f003:**
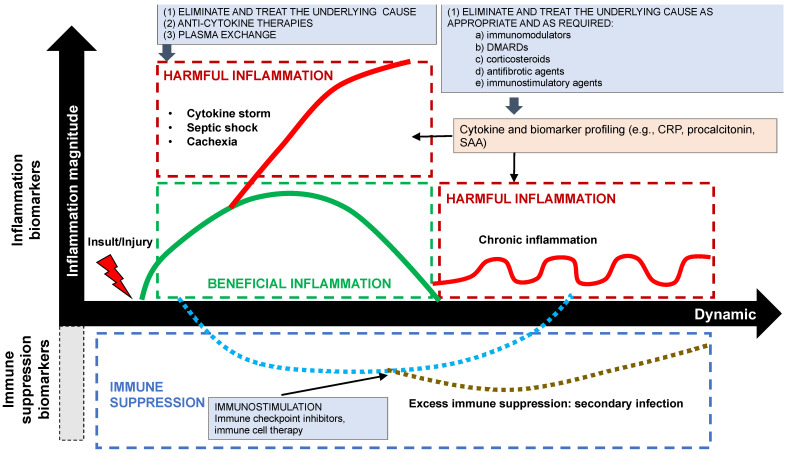
Potential Therapeutic Options Depending on Clinical Context Depending on Whether Hyperinflammation, Chronic Inflammation or Immunosuppression Is Present.

**Table 1 ijms-23-14905-t001:** Examples of Anti-inflammatory Immune Interventions.

	Antigen-Directed Immunotherapy	Immunomodulation	Anti-Fibrotic Agents	Immunosuppression
Diseases	asthma, allergic rhinitis, atopic dermatitis, anaphylaxis	SLE, RA, Sjögren’s, COPD, UC	pulmonary fibrosis	autoimmune diseases
Treatments	allergen immunotherapy	HCQ, sulfasalazine, dapsone, azithromycin	nintedanib, pirfenidone	corticosteroids, DMARDs, anti-cytokines
Treatment characteristics	potential for permanent desensitization	non-immunosuppressive	anti-fibrotic and anti-inflammatory effects especially with pirfenidone	more targeted immunosuppression with anti-cytokines

Abbreviations: SLE, systemic lupus erythematosus; RA, rheumatoid arthritis; COPD, chronic obstructive pulmonary disease; UC, ulcerative colitis; HCQ, hydroxychloroquine; DMARDs, disease-modifying anti-rheumatic drugs.

## Data Availability

Not applicable.

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
