# Peer review of "What Exactly Is Inflammation (and What Is It Not?)â€"

_ijms, 2022, doi:10.3390/ijms232314905_

Round 1
Reviewer 1 Report
This manuscript compared the onset and development of inflammation to a raging wildfire, from the cause of wildfire to fire accelerants, and then to put out. The explanation process was easy to understand, but the reviewers believe that some of the metaphors are unreasonable. The manuscript briefly mentioned the definition, mechanism, process, prevention, and treatment measures of inflammation, but lack of novelty. Other suggestions are listed below:
1. The full text was not closely relevant to the title, and inflammation is not clearly defined, focusing on the mechanism of inflammation and the treatment of different inflammations.
2. The abstract did not cover the whole content of the manuscript.
3. Line 68-76: the whole paragraph first described the stimuli of acute inflammation to chronic, followed by several clinical diseases that causes chronic inflammation, from molecular mechanisms to clinical cases. Such description is too simple.
4. Line 130-136: self-associated molecular patterns were described so briefly that the direct link of it to inflammation was not written. Ending with decreasing in oxidative stress, the link to inflammation is too broad and not detailed enough.
5. Line 160-166:a “watchful waiting” refers to the period of waiting for the inflammation to subside naturally before medication, but it does not indicate the length of the waiting process, the criteria for determination and the time for opportune medication. The use of the drugs(NSAIDs,corticosteroids, et al.)mentioned later was not explained in detail.
6. Line 168:the whole paragraph describes immune-directed therapies. More explanations should be given to connect it to inflammation.
7. Line 256-260: “it is not a pathological response …… These include arthritis, tendinitis, rhinitis, colitis, mucositis, cellulitis, bronchitis, esophagitis, appendicitis, cholecystitis, pancreatitis, meningitis, gingivitis etc.” Meningitis is a diffuse inflammation usually caused by bacteria or fungi. I don't understand why it is not a pathological response.
8. Line 267-274: in line 167-200, immune-directed therapies were described, such as the immunosuppressive agents, and treatment for the cause of disease was introduced in line 267-274, including antifungal and antiviral treatment. The relationship between the two parts was not clear.
9. Line 288-291:how cytokine indicates inflammatory therapy was not described, and more detailed examples should be given to support the argument.
10. Figure 3. The figure legend was too simple.
Author Response
Please see attachment. Our responses are in red. Thank you

Reviewer 2 Report
The review, or perspective, by Oronsky, Caroen, and Reid attempts to better define inflammation in order to identify better strategies for its management. The manuscript is generally well written and well organized and presented with a clear focus. The concluding section discusses different points on the beneficial and less beneficial consequences of, or opposite reactions to inflammation, and presents a graphical summary of the potential therapeutic options, depending on the clinical context. A slight concern for this reviewer is the low number of references for such a vast topic, but the authors argue in their introductory remarks that their intention is not to write an extensive review but rather formulate a perspective. The manuscript is therefore fairly superficial but it discusses several concepts that are likely to attract some interest in the current context of COVID and post-COVID management. More specific suggestions for improvement can be found below.
Specific comments
Line 6 – “first coined by the Romans, whose…” – please reformulate to avoid ambiguity. “Whose” is not referring to Romans.
Lines 22-28 – Please review punctuation re: different items on the list
Lines 40-46 – idem
Lines 50-52 – is the text in italics a direct quote or a definition coined by the authors? If the former, please add a reference.
Lines 78-91 – the authors include T cells as ‘arsonists’ in the section title but do not describe or discuss their role in immunopathogenesis and/or propagation of inflammation. This part should be expanded.
Lines 87-89 – efferocytosis is generally the work of M2 macrophages, not that of pro-inflammatory M1 macrophages. Efferocytic macrophages can be derived from M1 polarized cells, as inferred to later in text (lines 95-96), but this usually occurs at the resolution stage. The authors should consider rephrasing this section as it may lead to confusion.
Lines 108-113 – Does not appear to be a complete sentence. Please reformulate.
Fan the flames, lines 139-153 – I would invite the authors to expand a little on the timing of checkpoint inhibitors in sepsis. It is clear that reduced antigen presentation and lymphocyte apoptosis become an issue for secondary responses but suppressing the cytokine storm is essential for immediate survival in sepsis and hyperinflammation. Given that cytokine storm is initiated before the adaptive immune responses have kicked in, it is not immediately clear to me how checkpoint inhibitors can be of use in the early phase. This should be clarified.
Let it flame out, lines 163-166 – here, too, it would be helpful to expand on the idea and give more detailed examples of how anti-inflammatory drugs interfere with natural resolution
Prevent it, lines 203-208 – long sentence, please reformulate to avoid ambiguity (“with reduced visceral and ectopic fat accumulation” should probably be placed elsewhere)
Lines 221, 223 – using “anti-inflammatories” “anti-inflammatory” as noun – please replace with “anti-inflammatory therapies” or “anti-inflammatory agents / drugs”
Author Response
Please see attachment. Our responses are in red font. We thank the reviewer for the helpful comments

Round 2
Reviewer 1 Report
accept